# rhTPO Ameliorates Radiation-Induced Long-Term Hematopoietic Stem Cell Injury in Mice

**DOI:** 10.3390/molecules28041953

**Published:** 2023-02-18

**Authors:** Hao Luan, Jinkun Yang, Yemei Wang, Xing Shen, Xuewen Zhang, Zizhi Qiao, Shuang Xing, Zuyin Yu

**Affiliations:** 1School of Life Science, Anhui Medical University, Hefei 230032, China; 2Beijing Key Laboratory for Radiobiology, Department of Experimental Hematology and Biochemistry, Beijing Institute of Radiation Medicine, Beijing 100850, China; 3Graduate School, Jinzhou Medicine University, Jinzhou 121001, China

**Keywords:** ionizing radiation, hematopoietic stem cell, recombinant human thrombopoietin (rhTPO)

## Abstract

Exposure to medium and high doses of ionizing radiation (IR) can induce long-term bone marrow (BM) suppression. We previously showed that recombinant human thrombopoietin (rhTPO) significantly promotes recovery from hematopoietic-acute radiation syndrome, but its effect on long-term BM suppression remains unknown. C57BL/6 mice were exposed to 6.5 Gy γ-rays of total body irradiation (TBI) at a dose-rate of 63.01 cGy per minute, and the mice were treated with rhTPO (100 μg; intramuscular injection) or vehicle at 2 h after TBI. All mice were killed one or two months after TBI for analysis of peripheral blood cell counts, long-term hematopoietic stem cell (HSC) frequency, and BM-derived clonogenic activity. The HSC self-renewal capacity was analyzed by BM transplantation. The levels of reactive oxygen species (ROS) production and ratios of γH2AX+ and p16, p53, and p21 mRNA in HSCs were measured by flow cytometry and real-time polymerase chain reaction, respectively. Treatment with rhTPO reduced long-term myelosuppression by improving long-term hematopoietic reconstitution (*p* < 0.05) after transplantation and resting state maintenance of HSCs (*p* < 0.05). Moreover, rhTPO treatment was associated with a sustained reduction in long-term ROS production, reduction of long-term DNA damage, diminished p53/p21 mRNA expression, and prevention of senescence after TBI. This study suggests rhTPO is an effective agent for treating IR-induced long-term BM injury because it regulates hematopoietic remodeling and HSC cycle disorder through the ROS/p53/p21/p16 pathway long term after IR.

## 1. Introduction

Acute myelosuppression and long-term bone marrow (BM) damage may occur in individuals exposed to nuclear power plant leaks and nuclear war situations, as well as cancer patients receiving medium- and high-dose radiation [1,2]. Moreover, the hematopoietic system is highly sensitive to ionizing radiation (IR) [3]. After exposure to IR, hematopoietic cells suffer apoptosis, resulting in decreased hematological function, infection, bleeding, and anemia, which are the major causes of death in acute radiation sickness [4,5]. To quickly increase the number of mature hematopoietic cells after IR, thrombopoietin (TPO), granulocyte-macrophage colony-stimulating factor, granulocyte colony-stimulating factor, and other hematopoietic growth factors (HGFs) have become the most commonly used medications in the clinical treatment of acute myelosuppression [6]. However, the effects of HGF application in the acute phase after IR on long-term myelosuppression remain controversial, although the majority of these patients could recover after treatment [7].

After recovery from acute myelosuppression, although blood cell counts and BM cell numbers have usually returned to normal, residual BM injury is easily overlooked and could manifest as reduced hematopoietic stem cell (HSC) pool reserve and decreased self-renewal capacity [8,9,10]. Furthermore, studies have found that, while granulocyte colony-stimulating factor mobilizes HSC proliferation, it also accelerates the depletion of the HSC pool, delays adverse reactions, impairs the long-term recovery of BM hematopoietic function, and causes anemia, lymphoma, leukemia, and other diseases, thus negatively affecting patients’ quality of life [11,12,13]. Therefore, it is necessary to define the effects of HGF therapy administered in the acute phase after IR on long-term myelosuppression.

Recombinant human thrombopoietin (rhTPO), a drug with the same amino acid structure as endogenous TPO, promotes the proliferation of megakaryocytes [14]. It is widely used in the treatment of thrombocytopenia and immune thrombocytopenia caused by chemoradiotherapy [15,16]. Previous studies have demonstrated that TPO regulates platelet levels in the body, regulates the self-renewal of HSCs, maintains the resting state of HSCs, protects the genome stability of HSCs, and maintains homeostasis of the immune system [17,18,19,20,21]. Therefore, the application of TPO is a potentially effective countermeasure against radiation-induced myelosuppression. Our antecedent study demonstrated that a single injection of rhTPO 2 h after irradiation significantly improved early hematopoietic injury after irradiation by alleviating DNA damage and apoptosis, enhancing HSC proliferation, and effectively avoiding the neutralizing antibody response caused by multiple injections [22]. However, its effects on long-term myelosuppression and related mechanisms have not yet been reported.

In this study, we examined the effects of rhTPO on long-term BM damage in an assured model of IR-induced hematopoietic injury in mice. The results showed that rhTPO therapy could significantly alleviate the radiation-induced reduction of hematopoietic progenitor cell (HPC) clonal formation and hematopoietic reconstitution capacity, and maintain the self-renewal ability of HSCs. We further sought to elucidate the underlying molecular mechanisms associated with chronic oxidative stress and DNA damage in HSCs.

## 2. Result

### 2.1. rhTPO Increased Survival Rate of Mice Exposed to a Lethal Dose of Total Body Irradiation (TBI)

Our previous study showed that a single administration of rhTPO increased the probability of survival in lethally irradiated mice [22]. It showed that skin injection of rhTPO100 μg/kg 2 h after IR was the best administration scheme in the post-IR survival experiment. Based on this, we revisited the intramuscular injection administration route and found that it had a better therapeutic effect. Results showed that 100% of the mice died no more than 15 days after IR exposure (Figure 1A). The survival rates of the intramuscular and skin-injected group were 90% (7 out of 8) and 40% (3 out of 8), respectively. We then compared the effects of different doses of intramuscular injection. Single doses of 25, 50, and 100 μg/kg rhTPO resulted in the survival of 0% (0 out of 8), 50% (4 out of 8), and 90% (7 out of 8) of mice surviving for up to 30 days, confirming that 100 μg/kg is still the optimal administered dose. These results indicate that a single dose of rhTPO of 100 μg/kg, administered intramuscularly 2 h after IR, showed a significant ability to protect mice from lethal dose exposure.

### 2.2. rhTPO Promoted Long-Term Peripheral Blood Cell Recovery in Mice after TBI

Long-term myelosuppression is possible after TBI. With this condition, patients’ and animals’ blood cell count is usually normal despite a decrease in HSC reserves. To investigate how rhTPO treatment affects long-term hematopoietic recovery after irradiation, the number of peripheral blood cells was monitored for 60 days after 6.5 Gy TBI. Mice irradiated with 6.5 Gy γ-rays received intramuscular injections of rhTPO or saline on the basis of Figure 1B. Peripheral blood was extracted from the tail vein several times during two months after 6.5 Gy TBI, and the numbers of white blood cells (WBCs), platelets (PLTs), red blood cells (RBCs), and neutrophils (NEs) were measured. The results showed that rhTPO treatment significantly improved recovery from acute hematopoietic injury, and there were no differences in the numbers of RBCs, PLTs, WBCs, or NEs among any of the groups from 28 days after TBI (Figure 1B). This finding demonstrated that the long-term peripheral blood cell counts recovered after irradiation.

### 2.3. rhTPO Had No Significant Effect on the Phenotype of BM HSCs Long-Term after Irradiation

Long-term BM suppression manifested as a decrease in HSC reserves after recovery from acute myelosuppression. There were no differences in BMNC counts among any of the groups of mice at one or two months after IR (Figure 2A). The frequencies and numbers of different hematopoietic cell populations in BM cells were analyzed by flow cytometry (Figure 2B) to investigate the effects of rhTPO on the frequencies and numbers of HSCs (Lin−Scal+c-kit+), HPCs (Lin−Scal−c-kit+), LT-HSCs (CD34−CD135−LSK), ST-HSCs (CD34+CD135−LSK), and MPPs (CD34+CD135+LSK) after exposure to 6.5 Gy TBI. The results of this experiment showed that two months after TBI, the frequencies and numbers of HSCs, HPCs, ST-HSCs, and LT-HSCs in each group showed no significant difference (Figure 2C–J; Appendix A). The MPP ratio decreased significantly one and two months after irradiation, but rhTPO had no significant effect (Figure 2K,L). These results suggest that rhTPO did not affect the phenotype in different populations of hematopoietic cells two months after 6.5 Gy TBI.

### 2.4. rhTPO Mitigated TBI-Induced Long-Term Suppression of HPC Function

Previous studies have shown that IR impairs HPC function. Colony-forming unit (CFU) assays indicated that BM-derived clonogenic activity was markedly reduced in vehicle-treated mice one month after TBI. We examined whether the clonogenic activity of hematopoietic cells from IR mice was improved after rhTPO administration. As expected, rhTPO treatment significantly increased the CFU-E (17.3 ± 4.6 colonies vs. 8 ± 2 in vehicle; *p* < 0.05), BFU-E (12.7 ± 1.2 colonies vs. 2.7 ± 3.1 in vehicle; *p* < 0.01), and CFU-GM (32.0 ± 5.3 colonies vs. 14.0 ± 5.3 in vehicle; *p* < 0.05) (Figure 2M) at one month after TBI. Additionally, there were no significant differences in the clonogenic ability of HPCs among any of the groups at two months after irradiation (Figure 2N).

### 2.5. rhTPO Enhanced Long-Term and Multilineage Engraftment of Irradiated HSCs after BM Transplantation

Since long-term and multilineage engraftment is the current standard for measuring HSC function, we performed a competitive repopulation assay to investigate whether rhTPO treatment could improve HSC function. BM cells from sham-irradiated mice or irradiated mice treated with rhTPO or vehicle were mixed at a 20:1 ratio with CD45.1 competitor BM cells and injected into lethally irradiated CD45.1 recipients (Figure 3A). Either one month or two months after TBI for transplant assays, calculation of the percentage of peripheral blood chimerism every four weeks post-transplantation demonstrated a significantly decreased frequency of donor-derived cells in the recipients receiving BM cells from irradiated vehicle-treated donors compared with sham-irradiated donors (*p* < 0.001) (Figure 3B–D). Subsequently, the chimerism rate of rhTPO-treated BM cells was significantly higher than that of vehicle-treated BM cells (Figure 3B–D). Similarly, up to two months after irradiation, IR reduced the long-term contribution of HSCs to B cells and enhanced the myeloid cell ratio; however, rhTPO treatment improved the reestablishment of HSCs (Figure 3E,F). These findings suggested that rhTPO treatment can retain HSC function at two months after irradiation, thereby enhancing long-term and multilineage reconstitution after bone marrow transplantation.

### 2.6. rhTPO Attenuated TBI-Induced Long-Term Myelosuppression by Inhibiting HSC Proliferation

TBI may lead to apoptosis or cell cycle deregulation in HSCs. Thus, we probed whether rhTPO treatment could affect these cellular responses in HSCs after 6.5 Gy TBI. Cell-cycle state analysis showed that TBI significantly decreased quiescent cells in the LSK population, with 32.8% fewer LSK cells in the G0 phase of the cell cycle after vehicle treatment compared with the Normal group (*p* < 0.001) (Figure 4A,B). This was associated with the growing G1 and G2/S/M phase of LSKs one month after IR. As shown in Figure 4A,B, rhTPO treatment enhanced the self-renewal ability of HSCs one month after TBI. Additionally, these changes persisted only until one month after irradiation, and there were no significant changes in HSC cycles between groups two months after irradiation. Similarly, one month after IR, the apoptosis rate of HSCs in mice was dramatically increased (*p* < 0.05), but rhTPO administration could not relieve it. However, at two months after IR, the HSC cycle ratio and apoptosis rate of HSCs had no significant differences among any of the groups (Figure 4D,E).

### 2.7. rhTPO Inhibited Chronic Oxidative Stress and DNA Damage in HSCs Induced by IR

Previous studies have shown that the exposure of mice to sublethal doses of IR induces persistent hematopoietic oxidative stress leading to long-term BM suppression. As shown in Figure 5, the reactive oxygen species (ROS) production increased significantly until two months after IR. Congruously, rhTPO treatment attenuated the elevation in ROS production in HSCs. Subsequently, we examined whether rhTPO can reduce TBI-induced DNA damage in HSCs. This was achieved by flow cytometric analysis of γH2AX staining in HSCs, since γH2AX staining has been widely used as a surrogate for DNA double-strand breaks. Consistent with previous studies, our results also displayed increased production of γH2AX in HSCs one month after irradiation (Figure 5C,D), and γH2AX levels returned to normal two months after TBI. Treatment with rhTPO significantly reduced the increase in γH2AX staining in irradiated HSCs, illustrating that rhTPO treatment can inhibit TBI-induced DNA double-strand breaks in HSCs.

### 2.8. rhTPO Had the Ability to Moderate HSC Senescence Accompanying Long-Term Myelosuppression Induced by TBI

Mice exposed to TBI experience sustained oxidative stress and DNA damage in HSCs, leading to HSC senescence and long-term bone marrow suppression [23,24]. In our study, this is supported by the findings that HSCs from vehicle-treated TBI mice expressed elevated levels of p16 mRNA (Figure 5E), a widely used senescence biomarker and an important mediator of cellular senescence induction. In addition, after treatment with rhTPO, the IR mice displayed some recovery of HPC function (Figure 2) and remarkable improvement in hematopoietic reconstitution (Figure 3), indicating that rhTPO administration inhibited the IR-induced HSC senescence. This assumption is confirmed by the finding that rhTPO treatment also reduced the IR-induced expression of p16 mRNA in HSCs (Figure 5F). These findings indicate that treatment with rhTPO can attenuate TBI-induced residual BM injury in part via inhibition of HSC senescence.

When HSCs were in a state of oxidative stress, ROS activated the p53-p21 pathway by inducing DNA double-strand breaks, inducing HSC senescence. We detected the mRNA expression of p53 and p21 in HSCs one month after IR to explore the mechanism of rhTPO alleviating HSC senescence by reducing ROS. In our study, 6.5 Gy TBI resulted in a dramatic increase in p53 and p21 activation in HSCs (Lin−Scal+c-kit+ cells), and rhTPO administration reduced their expression. These results suggested that rhTPO alleviated IR-induced senescence in HSCs, in part via the ROS/p53/p21/p16 pathway.

## 3. Discussion

Currently, rhTPO is widely used in clinical thrombocytopenia as an HGF drug, and its radiation protection potential is gradually being explored [25]. Additionally, several studies demonstrated it brought significant improvements in acute radiation sickness [22,26,27]. However, whether rhTPO improves IR-induced long-term HSC function and, if so, the associated mechanism had not been elucidated. Hence, we first confirmed the optimal administration of rhTPO through survival experiments. On this basis, our study showed that rhTPO not only protected mice from acute myelosuppression but also ameliorated long-term BM damage induced by TBI.

Previous studies have shown that, one month after exposure to 4 Gy TBI, mice develop long-term BM suppression, with normal blood cell counts and HSC numbers. However, the clonal formation of HPCs is impaired, and the hematopoietic reconstitution function is decreased [11,24]. This phenomenon is consistent with our findings at one month after 6.5 Gy TBI. We added observations of various indicators in mice two months after IR and found that the proportion of MPPs decreased at both one and two months after IR, the hematopoietic reconstruction function of mice remained low at two months after IR, and the clonogenic capacity of HPCs had been basically restored. Interestingly, as with Li et al., mice that received donor cells from irradiated mice with vehicle treatment skewed toward more myeloid cell lineage in the competitive transplantation experiment, which is also one of the markers of hematopoietic aging [28]. Treatment with rhTPO both at one month and two months post-TBI showed stronger clonogenic capacity of HPCs and competitive hematopoietic reconstitution after transplantation, and corrected the myeloid cell skewing seen in the cells from the irradiated mice treated with vehicle after transplantation. This suggested that rhTPO may ameliorate IR-induced long-term HSC injury by inhibiting HSC senescence. 

The main cause of IR-induced long-term myelosuppression is the decreased self-renewal ability of HSCs, which is closely related to the regulation of the HSC cycle [1,29,30]. In our study, we first explored the cycle changes of HSCs in a 6.5 Gy TBI-induced long-term myelosuppression mouse model. The results showed that one month after IR, the fraction of HSCs in the G0 phase decreased, in the G1 phase arrested obviously, and in the G2/S/M phase increased. Two months after IR, the HSC cycle disorder was fundamentally restored to normal. Additionally, the use of HGF may aggravate long-term BM injury, mainly due to the reduction in the self-renewal ability of HSCs by HGF while promoting the proliferation and differentiation of HSCs and HPCs. This may lead to an accelerated depletion of HSCs and further jeopardize the long-term recovery of BM hematopoiesis. However, in our study, rhTPO treatment significantly ameliorated IR-induced HSC cycle changes but had no effect on HSC apoptosis. In previous studies, TPO^−/−^ mice showed decreased expressions of cell-cycle inhibitory molecules p57, p19, and Hoxb4 in the Hox family, increased BrdU+ HSC, and significantly decreased HSC in the G0 phase [18,31,32]; this suggested that rhTPO may improve the static maintenance of IR-induced HSC cells by regulating the expression of cell-cycle inhibitory molecules and a Hox family member.

Several studies have established that IR-induced chronic oxidative stress causes DNA damage and subsequently induces a decline in HSC self-renewal and aging, and that it is one of the dominant mechanisms of long-term BM suppression [23,24,33]. Further, several studies confirmed that the p53-p21 pathway, triggered by DNA damage, is the dominant pathway that induces the up-regulation of p16 expression in HSC senescence [28]. When DNA is damaged by ROS, ATM/ATR/DNA-PK/JNK are activated, leading to phosphorylation of the N-terminus of p53 by these kinases [34]. Then, the phosphorylation of p53 induces the increased expression of the downstream target p21, leading to cell-cycle arrest and senescence [35]. Likewise, our results suggest that rhTPO therapy also attenuates IR-induced chronic oxidative stress and DNA damage. Furthermore, rhTPO significantly downregulates p16 expression and reduces IR-induced HSC senescence, which may be related to the regulation of p53-p21 expression in HSCs after radiation.

Furthermore, compared with ST-HSC, LT-HSC could maintain long-term multilineage hematopoietic reconstruction and self-renewal [36,37]. Although rhTPO could significantly improve IR-induced long-term hematopoietic dysfunction and alleviate chronic oxidative stress and DNA damage in HSCs, whether rhTPO has a role in LT-HSC remains to be further investigated. Additionally, future studies could observe the changes in mice at longer time points and with more specific hematopoietic stem cell types after IR to fully explore the long-term myelosuppression caused by radiation.

The currently marketed drug of rhTPO is a fully glycosylated recombinant human thrombogenesis factor that is extracted and purified from Chinese hamster ovarian cells, and no neutralizing antibody has been found [14]. In addition to the hematopoietic system, rhTPO has a protective effect on multiple organs, such as the nerves, ovaries, and heart [38]. At present, rhTPO has been used clinically to treat chemotherapy-induced thrombocytopenia, but its effects on chemotherapy-induced long-term myelosuppression and specific adjuvant efficacy have not been studied [39]. Thus, it is of great significance to determine the efficacy and mechanism of rhTPO in the combination treatment of BM suppression in patients undergoing chemoradiotherapy or who have experienced nuclear events.

## 4. Methods and Materials

### 4.1. Reagents

Regarding the reagents used in this study, rhTPO was procured from Shenyang Sunshine Pharmaceutical Co., Ltd. (Shenyang, China). Anti-mouse CD117 (c-kit)-APC (clone2B8), anti-mouse Lineage-PercpCy5.5 (51-9006964), anti-mouse Ly-6 A/EA (Sca-1)-PE/Cy7 (cloneD7), anti-mouse CD135-PE (cloneA2F10), anti-mouse CD34-FITC (cloneRAM34), anti-mouse CD45.1-PE (cloneA20), anti-mouse CD45.2-FITC (clone104), anti-mouse CD45.1-APC, anti-mouse CD45.2-PECy7, anti-mouse B220-PE (cloneRA3-6B2), anti-mouse CD3-FITC (clone17A2), anti-mouse CD11b-PE (M1/70), anti-mouse Gr-1-FITC (cloneRB6-8C5), anti-mouse Ki67-PE (cloneSolA15), anti-mouse 7AAD-PercpCy5.5 (51-88981E), anti-mouse Annexin V-PE (640908), and anti-mouse γH2AX-FITC (clone2F3) were obtained from Biolegend (San Diego, CA, USA). Additionally, 2’, 7′-dichlorofluorescin diacetate (DCFDA) was purchased from Invitrogen (Carlsbad, CA, USA).

### 4.2. Mice

Male C57BL/6J (CD45.2) mice were obtained from Beijing HFK Bioscience (Beijing, China). B6.SJL/BoyJ (CD45.1) mice were purchased from Beijing Vital River Laboratory Animal Technology (Beijing, China). All mice were used in experiments at approximately 7–9 weeks of age. All procedures were approved by the Animal Care and Use Committee of the Animal Center in Academy (IACUC-DWZX-2022-836).

### 4.3. Irradiation and rhTPO Administration

Mice were exposed to a sublethal dose (6.5 Gy) or lethal dose (9.2 Gy) of TBI using a 60Co γ-ray source at a dose rate of 63.01 ± 0.5 cGy/min. For the optimal dosage study, C57BL/6J mice were subcutaneously given either phosphate-buffered saline (PBS) (vehicle; intramuscular injection), rhTPO (25 μg/kg; intramuscular injection), rhTPO (50 μg/kg; intramuscular injection), or rhTPO (100 μg/kg; intramuscular injection and subcutaneous injection) at 2 h after 9.0 Gy TBI. For the optimal administration-time study, mice were administered either PBS or 100 μg/kg rhTPO at 2, 12, and 24 h after 9.0 Gy TBI. Survival was checked and scored daily for 30 days. For the experiment, C57/BL mice were divided randomly into three groups: (A) Normal, (B) IR + vehicle, and (C) IR + rhTPO. The Normal mice were sham-irradiated. The IR + vehicle and IR + rhTPO groups were exposed to 6.5 Gy TBI and received PBS or rhTPO (100 μg/kg; intramuscular injection), respectively.

### 4.4. Peripheral Blood Cell and BM Nucleated Cell (BMNCs) Counts

The mice were kept in narcosis, and then blood was obtained through the caudal vein. The numbers of peripheral blood cells, including WBCs, PLTs, RBCs, and NEs, were measured by an MEK-7222K (Nihon kohden, Tokyo, Japan) before IR and at 10, 14, 18, 21, 28, and 60 days post-TBI. The BMNC counts were determined as we previously reported.

### 4.5. Hematopoietic Cell Phenotypic Analysis by Flow Cytometry

Briefly, 1 × 10^7^ BM cells were incubated with lineage antibodies (CD4, CD8, CD45R/B220, Gr-1, Mac-1, and Ter-119), and then the cells were washed with PBS and reacted with anti-CD16/CD32 antibody to block Fc receptors. After that, the frequencies of HPCs (Lin−Sca1−c-kit+ cells), LSK cells (Lin−Sca1+c-kit+ cells), short-term HSCs (ST-HSCs, CD34+ LSK cells), long-term HSCs (LT-HSCs, CD34−CD135−LSK cells), multipotent progenitors (MPPs, CD34+CD135+LSK cells) were analyzed with a flow cytometer (Thermo Fisher Scientific, Waltham, MA, USA).

### 4.6. BM-Derived Clonogenic Activity Assay

One and two months after IR, 1 × 10^5^ BM cells were collected from each group and cultured for colony-forming units in MethoCult medium (StemCell Technologies, Vancouver, Canada). Then, the colonies were counted on day 7, as described previously.

### 4.7. Competitive Repopulation Assay

Briefly, BM cells were obtained from donor C57/BL (CD45.2) mice after the various treatments, and 2 × 10^6^ of the BM cells were mixed with 1 × 10^5^ competitive BM cells (CD45.2&CD45.1). The mixed cells were transplanted into lethal IR (9.2 Gy) B6.SJL mice (CD45.1) by tail vein injection. To analyze the donor cell engraftment, peripheral blood was collected at 4, 8, 12, and 16 weeks after transplantation and stained with anti-CD45.1-PE and CD45.2-FITC. In week 12, the peripheral blood was stained with anti-CD45.1-APC, CD45.2-PECy7, B220-PE, CD11b-PE, Gr-1-FITC, and CD3-FITC. The stained blood cells were analyzed using a flow cytometer (Thermo Fisher Scientific, USA).

### 4.8. Analysis of the Apoptosis in HSCs

BM cells were flushed from mouse femurs with RPMI-1640 medium containing 2% fetal bovine serum and lysed using a lysis buffer (StemCell Technologies). Then 1 × 10^7^ of the BM cells were stained with LSK, as described previously. The cells were incubated with anti-Annexin V-PE and anti-7AAD. The cells were tested using a flow cytometer (Thermo Fisher Scientific, USA), and the flow cytometry data were analyzed using FlowJo 7.6.1 (Tree Star, Franklin Lake, NJ, USA).

### 4.9. Analysis of the Proliferation in HSCs

First, 1 × 10^7^ BM cells were stained with LSK, as described previously. Then, the cells were fixed and permeabilized by BD Cytofix/Cytoperm buffer according to the manufacturer’s protocol and incubated with anti-Ki67-PE and anti-7AAD. Finally, the cells were tested using a flow cytometer (Thermo Fisher Scientific, USA), and the flow cytometry data were analyzed using the FlowJo 7.6.1 software (Tree Star, USA).

### 4.10. Analysis of the Levels of ROS in HSCs

BM cells were flushed from mouse femurs with RPMI-1640 medium containing 2% fetal bovine serum and lysed using a lysis buffer (StemCell Technologies). Then, 1 × 10^7^ of the BM cells were stained with LSK, as described previously. Next, the cells were incubated with DCF (10 μM) for 30 min at 37 °C. After washing with PBS, the cells were analyzed using a flow cytometer (Thermo Fisher Scientific, USA).

### 4.11. Analysis of γH2AX Staining in HSCs

First, 1 × 10^7^ BM cells were stained with LSK, as described previously. Then, the cells were fixed and permeabilized, as described above, and incubated with anti-γH2AX-FITC. After washing with Buffer (Invitrogen, Carlsbad, CA, USA), the cells were analyzed on a flow cytometer (Thermo Fisher Scientific, USA). 

### 4.12. Quantitative real-Time Polymerase Chain Reaction (PCR) Assays

BM cells were incubated with lineage beads, and then cells binding the paramagnetic beads were removed with a magnetic field. Lin− cells were stained with LSK as described above, and LSK+ cells (Lin−Sca1+c-kit+ cells) were sorted using a FACStarPlus cell sorter (Becton Dickinson, Franklin Lake, NJ, USA). Then, total RNA and cDNA were obtained from 10,000 sorted HSCs using a Single Cell Sequence Specific Amplification Kit (Vazyme Biotech, Nanjing, China) following the manufacturer’s protocol. The PCR primers for p16, p53, p21, p38, and GAPDH were obtained from Sangon Biotech (Shanghai, China). The cDNA samples were mixed with primers and SYBR Master Mix (Vazyme Biotech, China) in a total volume of 20 mL. All samples were analyzed in triplicate using a CFX96TM Real-Time System (Bio-Rad, Hercules, CA, USA). The threshold cycle (CT) values for each reaction were determined and averaged using TaqMan SDS analysis software (Applied Biosystems 2.1). Changes in gene expression were calculated by the comparative CT method (fold changes = 2^−ΔΔCT^).

### 4.13. Statistical Analysis

All results were expressed as the mean ± standard deviation (SD). One-way analysis of variance was used for multiple comparisons. Survival was evaluated by Kaplan–Meier analysis with the log-rank test. All analyses were performed using GraphPad Prism 8.0 (San Diego, CA, USA). Statistical significance was defined as *p* < 0.05.

## 5. Conclusions

This study suggests that rhTPO is an effective agent for IR therapy because it improves not only acute myelosuppression but also long-term BM injury. Additionally, rhTPO could regulate hematopoietic remodeling and HSC cycle disorder through the ROS/p53/p21/p16 pathway long term after IR, thereby ameliorating IR-induced long-term myelosuppression in mice.

## Figures and Tables

**Figure 1 molecules-28-01953-f001:**
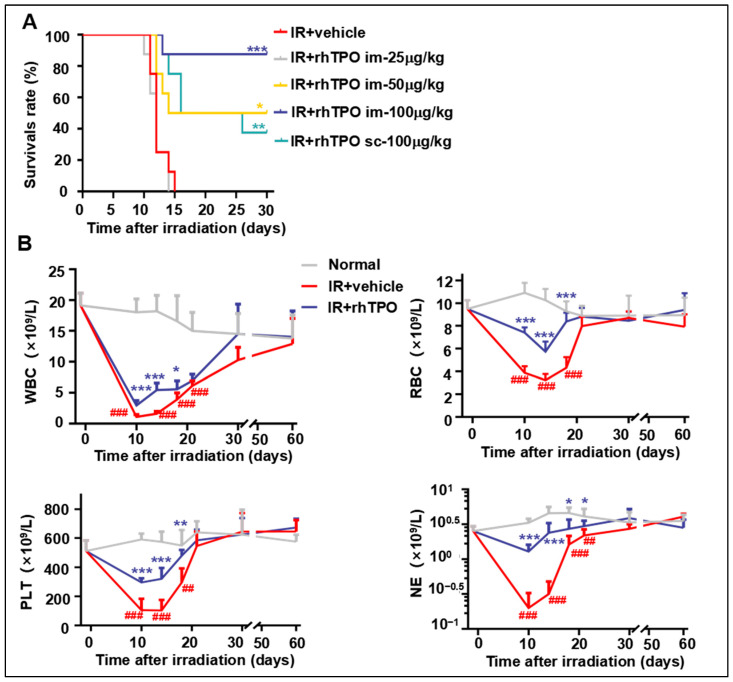
Effects of recombinant human thrombopoietin on survival and hematopoietic cell recovery in irradiated mice. (**A**) Survival of C57BL/6J mice was monitored for 30 days after 9.2 Gy TBI. Kaplan–Meier survival rate curves of 5 groups (*n* = 8 per group) administered with phosphate-buffered saline (PBS; vehicle) or indicated doses of rhTPO 2 h after IR. (**B**) Peripheral blood cell counts of white blood cells (WBCs), red blood cells (RBCs), platelets (PLTs), and neutrophils (NEs) of mice (*n* = 6 per group) administered with PBS (vehicle) or rhTPO (100 μg/kg) 2 h after IR. Data are reported as the mean ± standard deviation. * *p* < 0.05, ** *p* < 0.01, *** *p* < 0.001 versus the normal group. ^##^
*p* < 0.01, ^###^
*p* < 0.001 versus irradiation treated with the vehicle group.

**Figure 2 molecules-28-01953-f002:**
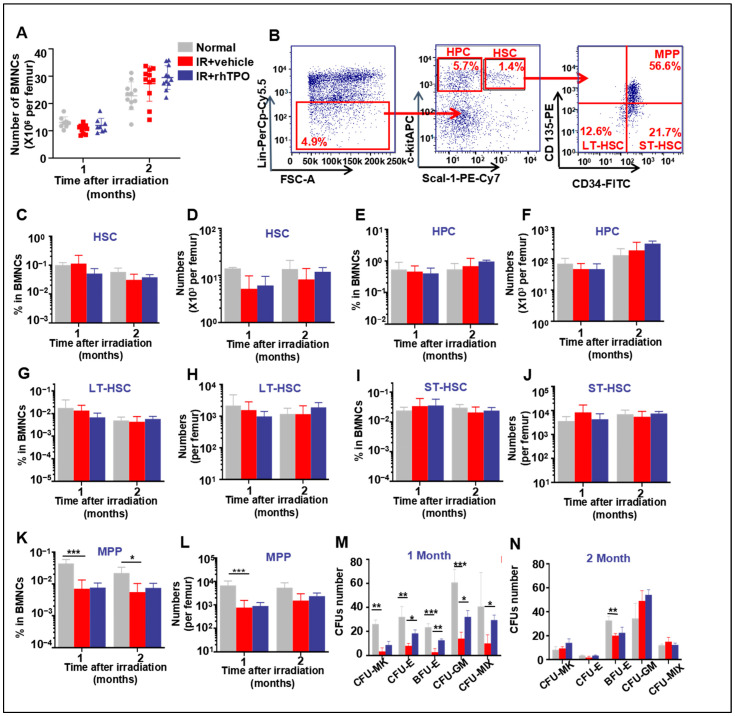
The influence of rhTPO on the frequencies and numbers of long-term hematopoietic cells and BMNCs and on the BM colony-forming ability one and two months after TBI. Sham-irradiated mice were administered phosphate-buffered saline (PBS) and shielded from irradiation. The vehicle and rhTPO groups were administered PBS or 100 μg/kg rhTPO, respectively, at 2 h after 6.5 Gy TBI (*n* = 6 per group). BM cells were harvested at one and two months post-irradiation. (**A**) Bone marrow mononuclear cell (BMNC) count. (**B**) A representative gating strategy of LT-HSC, ST-HSC, and MPP was analyzed by flow cytometry. The frequencies of (**C**) HSC (Lin−Scal+c-kit+), (**E**) HPC (Lin−Scal−c-kit+), (**G**) LT-HSC (CD34−CD135−LSK), (**I**) ST-HSC (CD34+CD135−LSK), and (**K**) MPP (CD34+CD135+LSK) in BMNCs (%). The numbers of (**D**) HSCs, (**F**) HPC, (**H**) LT-HSCs, (**J**) ST-HSCs, and (**L**) MPPs per mouse. Colony-forming units of the mice (*n* = 5 per group) treated with PBS or rhTPO at (**M**) one month and (**N**) two months after 6.5 Gy TBI. Data are reported as the mean ± standard deviation. * *p* < 0.05, ** *p* < 0.01, *** *p* < 0.001.

**Figure 3 molecules-28-01953-f003:**
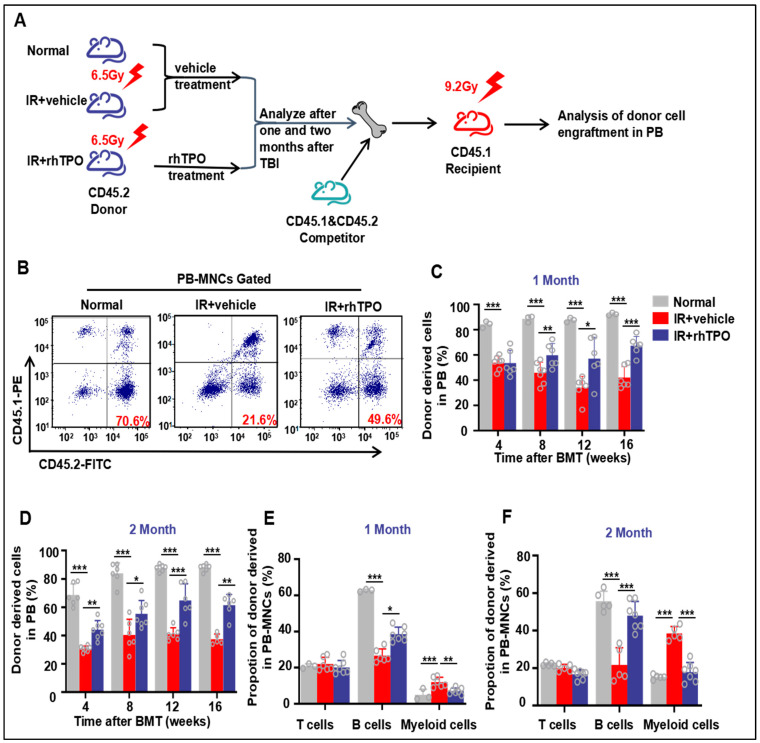
Injection of rhTPO enhances long-term and multilineage engraftment of irradiated HSCs after BM transplantation. The mice were sham-irradiated as a Normal control or subjected to IR and then treated with vehicle or rhTPO as described in the text. (**A**) Schematic of the experimental procedure. (**B**) Representative FACS plots and (**C**,**D**) percentages of donor (CD45.2) chimeras from Normal, IR + vehicle, and IR + rhTPO group mice in peripheral blood of recipient mice (CD45.1) (*n* = 6 per group) at 4, 8, 12, and 16 weeks after transplantation. (**E**,**F**) Frequencies of Normal, IR + vehicle, and IR + rhTPO group donor (CD45.2)-derived myeloid cells (CD45.2+ CD11b+ and/or Gr-1+) and T (CD45.2+CD3+) and B (CD45.2+ B220+) lymphocytes 12 weeks after transplantation in peripheral blood of recipient mice (CD45.1). Data are reported as the mean ± standard deviation. * *p* < 0.05, ** *p* < 0.01, *** *p* < 0.001.

**Figure 4 molecules-28-01953-f004:**
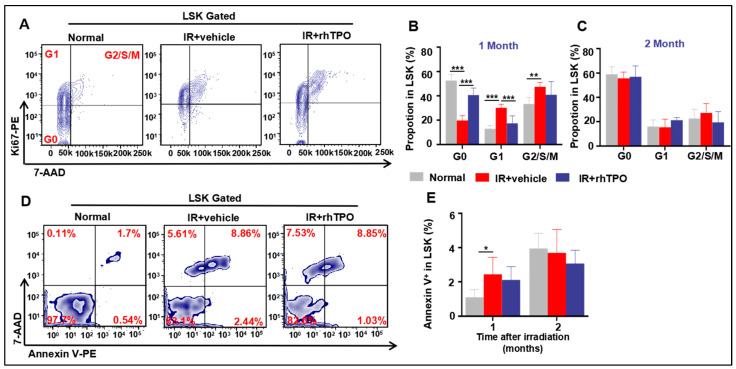
Injection of rhTPO attenuates TBI-induced residual BM injury in part via inhibition of HSC proliferation. Mice were sham-irradiated as controls or subjected to IR and then treated with vehicle or rhTPO one and two months after TBI as described above. (**A**) Representative FACS plots of HSC proliferation. The frequency of cell-cycle distribution (Ki67/7-AAD staining) in LSK cells was analyzed by flow cytometry (**B**) one month and (**C**) two months after TBI. (**D**) Representative FACS plots of HSC apoptosis and (**E**) the frequency of cell apoptosis (Annexin V/7-AAD staining) in LSK cells were analyzed by flow cytometry. Data are reported as the mean ± standard deviation (*n* = 5 per group). * *p* < 0.05, ** *p* < 0.01, *** *p* < 0.001.

**Figure 5 molecules-28-01953-f005:**
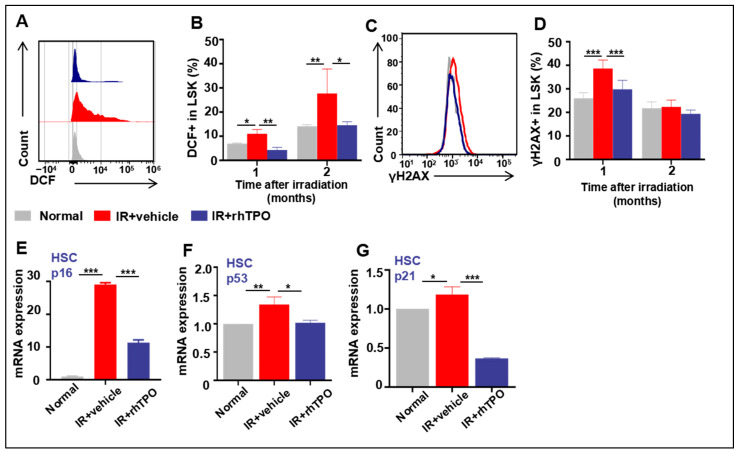
Effects of rhTPO on ROS, DNA damage, and cell senescence in HSCs after irradiation. Mice were sham-irradiated as controls or subjected to IR and then treated with vehicle or rhTPO as described above. The mice were euthanized one and two months after exposure to TBI to harvest BMNCs. (**A**) Representative analysis of ROS levels in HSCs by flow cytometry. (**B**) Percentages of DCF+ cells in LSKs from non-IR, vehicle, or rhTPO-treated mice (*n* = 5 per group), presented as the mean ± standard error. (**C**) Representative analysis of γH2AX expression in HSCs by flow cytometry. (**D**) Percentages of γH2AX+ cells in LSKs from non-IR, vehicle, or rhTPO-treated mice (*n* = 5 per group) were presented as the mean ± standard error. HSCs were isolated from BMNCs by cell sorting and analyzed for the expression of p16, p53, and p21 mRNA by real-time PCR. (**E**–**G**) Expression levels of p16, p53, and p21 mRNA in HSCs. The levels of p16, p53, and p21mRNA expression are expressed as the mean ± standard error of fold changes compared to the control (*n* = 8). * *p* < 0.05, ** *p* < 0.01, *** *p* < 0.001.

## Data Availability

Research data are stored in an institutional repository and will be shared upon request to the corresponding author.

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
