# Peer review of "rhTPO Ameliorates Radiation-Induced Long-Term Hematopoietic Stem Cell Injury in Mice"

_molecules, 2023, doi:10.3390/molecules28041953_

Round 1
Reviewer 1 Report
In this study, the authors investigated the effects of recombinant human thrombopoietin (rhTPO) in alleviating radiation-induced injury to long-term bone marrow injury. The authors show that rhTPO confers significant survival benefits against lethal dose of 9.2 Gy TBI and improved recovery of peripheral blood cells at early time points post sub-lethal TBI, but these differences were obliterated by 1 to 2 months post sub-lethal irradiation. Further, there were no differences observed in the recovery of peripheral blood cells or HSCs at 1mth and 2 mths post-TBI (Fig. 1, 2). These results suggest that TPO may have no effects on long-term bone marrow injury. The only differential effects that were observed were on the differentiation of progenitors and competitive repopulation assay which showed improved recovery of B-cells but not myeloid cells in mice cells from rhTPO treated mice.
Following points need to be addressed to substantiate that rhTPO indeed has an effect on long term bone marrow injury:
1. Have the authors examined the profile of LT-HSCs or ST-HSCs at later time points beyond 2 months in the bone marrow?
2. In Fig. 2K and 2L are labeled as MPP. Kindly specify which type of MPP, eg. MPP1 or MPP2 etc. Additionally, please provide a representative flow cytometry panel with all the populations as a supplementary file.
3. Bone marrow transplantation from rhTPO-treated mice both at 1 month and 2 months post-TBI showed significant increases in donor-derived B cells but not myeloid cells in CD45.1 recipient mice. Since myelosuppression is a hallmark of radiation injury to the bone marrow, it is interesting that rhTPO treatment does not alleviate myelosuppression, but promotes recovery of B-lymphocytes. This should be highlighted in the discussion section.
4. In order to correlate the finding of competitive repopulation results with changes in ROS, DNA damage, cell cycle. These should be examined in ST-HSCs versus LT-HSCs. If the same findings are observed in LT-HSCs, then this could support the statement that rhTPO alleviates long-term bone marrow injury.
Author Response
6th February, 2023
Professors
Section Managing Editor
Molecules
Manuscript ID: molecules-2174091
Manuscript Title: rhTPO ameliorates radiation-induced long-term hematopoietic stem cell injury in mice
Dear Editor,
Thank you for your letter in 21 January, 2023. On behalf of my co-authors, we thank you very much for giving us an opportunity to revise our manuscript, we appreciate editor and reviewers very much for their positive and constructive comments and suggestions on our manuscript.
Appended to this letter is our point-by-point response to the comments raised by the reviewer. The comments are reproduced and our responses are given directly afterward in a different color(highlight yellow).
We would like to express our great appreciation to you and reviewers for comments on our paper. We hope that we have responded satisfactorily to the constructive comments of the reviewers, and the paper will now be suitable for publication in the Molecules.
Best wishes,
The authors of paper molecules-2174091.
Response to Reviewer 1 comments
Thank you for your letter and for the reviewers’ comments concerning our manuscript entitled “rhTPO ameliorates radiation-induced long-term hematopoietic stem cell injury in mice” (Manuscript ID: molecules-2174091). Those comments are all valuable and very helpful for revising and improving our paper, as well as the important guiding significance to our researches. We have studied comments carefully and have made correction which we hope meet with approval. Revised portion are marked in highlight yellow(normal revision) in the paper. The main corrections in the paper and the responds to the reviewer’s comments are as flowing:
Point 1: In this study, the authors investigated the effects of recombinant human thrombopoietin (rhTPO) in alleviating radiation-induced injury to long-term bone marrow injury. The authors show that rhTPO confers significant survival benefits against lethal dose of 9.2 Gy TBI and improved recovery of peripheral blood cells at early time points post sub-lethal TBI, but these differences were obliterated by 1 to 2 months post sub-lethal irradiation. Further, there were no differences observed in the recovery of peripheral blood cells or HSCs at 1mth and 2 mths post-TBI (Fig. 1, 2). These results suggest that TPO may have no effects on long-term bone marrow injury. The only differential effects that were observed were on the differentiation of progenitors and competitive repopulation assay which showed improved recovery of B-cells but not myeloid cells in mice cells from rhTPO treated mice.
Following points need to be addressed to substantiate that rhTPO indeed has an effect on long term bone marrow injury:
Have the authors examined the profile of LT-HSCs or ST-HSCs at later time points beyond 2 months in the bone marrow?
Response 1: Thank you for this positive comment. This is exactly the future research direction, and we have already begun work in this area. At present, the research has not been completed due delays caused by the COVID-19 epidemic. You will see the research in this area in our future work. Recent research on drugs for the treatment of IR-induced long-term myelosuppression mainly focused on time points of one to two months after IR [1-4]. The detection of LT-HSC or ST-HSC at time points beyond two months after irradiation will be of great value for exploring the effects of irradiation and rhTPO on long-term BM suppression. We have added this information to the "Discussion" section of the resubmitted manuscript (please see lines 309-311, page 11).
Moreover, IR-induced long-term myelosuppression is potential, and peripheral blood cell counts and hematopoietic homeostasis are normalized in patients and animals after recovery from hematopoietic acute radiation sickness[5-6]. In addition, the use of hematopoietic growth factors (HGFs), such as G-CSF, may aggravate IR-induced long-term BM suppression, because HGFs reduce the self-renewal ability of HSCs while promoting the proliferation and differentiation of HSCs [7-8]. Our study innovatively explored the effects of rhTPO on IR-induced long-term myelosuppression for the first time. It was clear that rhTPO did not inhibit long-term BM hematopoietic homeostysis after IR, but it could improve the differentiation and hematopoietic reconstruction ability of long-term HPCs after radiation in mice. It is suggested that rhTPO could improve long-term radiation-induced myelosuppression. Additionally, several studies have demonstrated that IR-induced long-term myelosuppression mainly causes chronic oxidative stress and DNA damage in HSCs (Lin-Scal+c-kit+) [9-11]. Our experimental study found that rhTPO significantly improved the ROS and DNA damage in HSCs (Lin-Scal+c-kit+).
1. Guan B, Li C, Yang Y, Lu Y, Sun Y, Su L, Shi G, Bai L, Liu J, Meng A. Effect of spermidine on radiation-induced long-term bone marrow cell injury. Int Immunopharmacol. 2023 Jan;114:109557. doi: 10.1016/j.intimp.
2. Sim HJ, Bhattarai G, Lee J, Lee JC, Kook SH. The Long-lasting Radioprotective Effect of Caffeic Acid in Mice Exposed to Total Body Irradiation by Modulating Reactive Oxygen Species Generation and Hematopoietic Stem Cell Senescence-Accompanied Long-term Residual Bone Marrow Injury. Aging Dis. 2019 Dec 1;10(6):1320-1327. doi: 10.14336/AD.2019.0208.
3. Shao L, Feng W, Li H, Gardner D, Luo Y, Wang Y, Liu L, Meng A, Sharpless NE, Zhou D. Total body irradiation causes long-term mouse BM injury via induction of HSC premature senescence in an Ink4a- and Arf-independent manner. Blood. 2014 May 15;123(20):3105-15. doi: 10.1182/blood-2013-07-515619.
4. Lu L, Wang YY, Zhang JL, Li DG, Meng AM. p38 MAPK Inhibitor Insufficiently Attenuates HSC Senescence Administered Long-Term after 6 Gy Total Body Irradiation in Mice. Int J Mol Sci. 2016 Jun 8;17(6):905. doi: 10.3390/ijms17060905.
5. Wang Y, Schulte BA, LaRue AC, Ogawa M, Zhou D. Total body irradiation selectively induces murine hematopoietic stem cell senescence. Blood. 2006 Jan 1;107(1):358-66. doi: 10.1182/blood-2005-04-1418.
6. Meng A, Wang Y, Van Zant G, Zhou D. Ionizing radiation and busulfan induce premature senescence in murine bone marrow hematopoietic cells. Cancer Res. 2003 Sep 1;63(17):5414-9. PMID: 14500376.
7. Li C, Lu L, Zhang J, Huang S, Xing Y, Zhao M, Zhou D, Li D, Meng A. Granulocyte colony-stimulating factor exacerbates hematopoietic stem cell injury after irradiation. Cell Biosci. 2015 Nov 25;5:65. doi: 10.1186/s13578-015-0057-3.
8. Singh VK, Seed TM. Pharmacological management of ionizing radiation injuries: current and prospective agents and targeted organ systems. Expert Opin Pharmacother. 2020 Feb;21(3):317-337. doi: 10.1080/14656566.2019.1702968.
9. Shao L, Luo Y, Zhou D. Hematopoietic stem cell injury induced by ionizing radiation. Antioxid Redox Signal. 2014 Mar 20;20(9):1447-62. doi: 10.1089/ars.2013.5635.
10. Li H, Wang Y, Pazhanisamy SK, Shao L, Batinic-Haberle I, Meng A, Zhou D. Mn(III) meso-tetrakis-(N-ethylpyridinium-2-yl) porphyrin mitigates total body irradiation-induced long-term bone marrow suppression. Free Radic Biol Med. 2011 Jul 1;51(1):30-7. doi: 10.1016/j.freeradbiomed.
11. Wang Y, Schulte BA, Zhou D. Hematopoietic stem cell senescence and long-term bone marrow injury. Cell Cycle. 2006 Jan;5(1):35-8. doi: 10.4161/cc.5.1.2280.
Point 2: In Fig. 2K and 2L are labeled as MPP. Kindly specify which type of MPP, eg. MPP1 or MPP2 etc. Additionally, please provide a representative flow cytometry panel with all the populations as a supplementary file.
Response 2: We thank the reviewer for raising this question. According to their suggestion, we have added a representative flow cytometry panel with all the populations as a supplementary file.
Recently, multipotent progenitors (MPPs) have been divided into six groups: MPP1 (LSK, CD34+, CD48−, CD150+, CD135−), MPP2 (LSK, CD34+, CD48+, CD150+, CD135−), MPP3 (LSK, CD34+, CD48+, CD150−, CD135−), MPP4 (LSK, CD34+, CD48+, CD150−, CD135+), MPP5 (LSK, CD34+, CD48−, CD150−,CD135−), and MPP6 (LSK, CD34−, CD48−, CD150−, CD135−) [1-2]. In our study, we detected the MPPs (LSK, CD34+, CD135+) via flow cytometry. We regret that we did not delve into the specific types of MPP, and we agree that this is a potential limitation of this study. In this study, we mainly focused on the result of HSC types, referring to other IR-induced myelosuppression articles [3-4]. In addition, we would love to consider this point in follow-up research and add this to the "Discussion" section of the resubmitted manuscript (please see lines 309-311, page 11).
1. Wilson A, Laurenti E, Oser G, van der Wath RC, Blanco-Bose W, Jaworski M, Offner S, Dunant CF, Eshkind L, Bockamp E, Lió P, Macdonald HR, Trumpp A. Hematopoietic stem cells reversibly switch from dormancy to self-renewal during homeostasis and repair. Cell. 2008 Dec 12;135(6):1118-29. doi: 10.1016/j.cell.2008.10.048.
2. Sommerkamp P, Romero-Mulero MC, Narr A, Ladel L, Hustin L, Schönberger K, Renders S, Altamura S, Zeisberger P, Jäcklein K, Klimmeck D, Rodriguez-Fraticelli A, Camargo FD, Perié L, Trumpp A, Cabezas-Wallscheid N. Mouse multipotent progenitor 5 cells are located at the interphase between hematopoietic stem and progenitor cells. Blood. 2021 Jun 10;137(23):3218-3224. doi: 10.1182/blood.2020007876.
3. Li C, Lu L, Zhang J, Huang S, Xing Y, Zhao M, Zhou D, Li D, Meng A. Granulocyte colony-stimulating factor exacerbates hematopoietic stem cell injury after irradiation. Cell Biosci. 2015 Nov 25;5:65. doi: 10.1186/s13578-015-0057-3.
4. Xing S, Shen X, Yang JK, Wang XR, Ou HL, Zhang XW, Xiong GL, Shan YJ, Cong YW, Luo QL, Yu ZY. Single-Dose Administration of Recombinant Human Thrombopoietin Mitigates Total Body Irradiation-Induced Hematopoietic System Injury in Mice and Nonhuman Primates. Int J Radiat Oncol Biol Phys. 2020 Dec 1;108(5):1357-1367. doi: 10.1016/j.ijrobp.
Point 3: Bone marrow transplantation from rhTPO-treated mice both at 1 month and 2 months post-TBI showed significant increases in donor-derived B cells but not myeloid cells in CD45.1 recipient mice. Since myelosuppression is a hallmark of radiation injury to the bone marrow, it is interesting that rhTPO treatment does not alleviate myelosuppression, but promotes recovery of B-lymphocytes. This should be highlighted in the discussion section.
Response 3: We appreciate the reviewer for this valuable recommendation. The effect of rhTPO on B lymphocyte recovery after bone marrow transplantation in mice needs further exploration. It is a very worthwhile research topic. In addition, the reduction of myeloid cells in rhTPO-treated mice after bone marrow transplantation suggests that rhTPO could correct the IR-induced myeloid differentiation bias in HSCs. Accordingly, we have conducted an in-depth review and analysis of this finding and highlighted the results in the discussion section of the manuscript (please see lines 268-274, page 10).
As with Li et al., mice that received donor cells from irradiated mice with vehicle treatment skewed toward more myeloid cell lineage in the competitive transplantation experiment [1]. Further, mice developed long-term myeloid preference hematopoiesis after irradiation, suggesting impaired function of HSCs, which is also one of the markers of hematopoietic aging in mice [2-6]. It suggested that rhTPO may ameliorate IR-induced long-term HSC injury by inhibiting HSC senescence.
1. Li H, Wang Y, Pazhanisamy SK, Shao L, Batinic-Haberle I, Meng A, Zhou D. Mn(III) meso-tetrakis-(N-ethylpyridinium-2-yl) porphyrin mitigates total body irradiation-induced long-term bone marrow suppression. Free Radic Biol Med. 2011 Jul 1;51(1):30-7. doi: 10.1016/j.freeradbiomed.2011.04.016.
2. Bogeska R, Mikecin AM, Kaschutnig P, Fawaz M, Büchler-Schäff M, Le D, Ganuza M, Vollmer A, Paffenholz SV, Asada N, Rodriguez-Correa E, Frauhammer F, Buettner F, Ball M, Knoch J, Stäble S, Walter D, Petri A, Carreño-Gonzalez MJ, Wagner V, Brors B, Haas S, Lipka DB, Essers MAG, Weru V, Holland-Letz T, Mallm JP, Rippe K, Krämer S, Schlesner M, McKinney Freeman S, Florian MC, King KY, Frenette PS, Rieger MA, Milsom MD. Inflammatory exposure drives long-lived impairment of hematopoietic stem cell self-renewal activity and accelerated aging. Cell Stem Cell. 2022 Aug 4;29(8):1273-1284.e8. doi: 10.1016/j.stem.2022.06.012.
3. Montazersaheb S, Ehsani A, Fathi E, Farahzadi R. Cellular and Molecular Mechanisms Involved in Hematopoietic Stem Cell Aging as a Clinical Prospect. Oxid Med Cell Longev. 2022 Apr 1;2022:2713483. doi: 10.1155/2022/2713483.
4. Beerman I, Bhattacharya D, Zandi S, Sigvardsson M, Weissman IL, Bryder D, Rossi DJ. Functionally distinct hematopoietic stem cells modulate hematopoietic lineage potential during aging by a mechanism of clonal expansion. Proc Natl Acad Sci U S A. 2010 Mar 23;107(12):5465-70. doi: 10.1073/pnas.1000834107.
5. Dykstra B, Olthof S, Schreuder J, Ritsema M, de Haan G. Clonal analysis reveals multiple functional defects of aged murine hematopoietic stem cells. J Exp Med. 2011 Dec 19;208(13):2691-703. doi: 10.1084/jem.20111490.
6. Wang J, Geiger H, Rudolph KL. Immunoaging induced by hematopoietic stem cell aging. Curr Opin Immunol. 2011 Aug;23(4):532-6. doi: 10.1016/j.coi.2011.05.004.
Point 4: In order to correlate the finding of competitive repopulation results with changes in ROS, DNA damage, cell cycle. These should be examined in ST-HSCs versus LT-HSCs. If the same findings are observed in LT-HSCs, then this could support the statement that rhTPO alleviates long-term bone marrow injury.
Response 4: Thank you very much for pointing out this important issue. We designed the LSK-based study referring to the articles stating that drugs promote the recovery of IR-induced long-term myelosuppression in mice [1-4]. To some extent, the results of these studies suggest that rhTPO could regulate IR-induced long-term myelosuppression by ameliorating chronic oxidative stress and other mechanisms. We should indeed detect ROS and DNA damage-related changes in LT-HSC and ST-HSC to further clarify the mechanism of action of rhTPO on IR-induced long-term myelosuppression. We acknowledge that further analysis of the changes in LT-HSC would be more accurate and convincing. Unfortunately, due to the COVID-19 epidemic and limited time, we did not add experimental validation. This important issue will be studied in future research. Additionally, we have added this limitation to the “Discussion” section (please see lines 305-311, page 11).
1. Guan B, Li C, Yang Y, Lu Y, Sun Y, Su L, Shi G, Bai L, Liu J, Meng A. Effect of spermidine on radiation-induced long-term bone marrow cell injury. Int Immunopharmacol. 2023 Jan;114:109557. doi: 10.1016/j.intimp.2022.109557.
2. Sim HJ, Bhattarai G, Lee J, Lee JC, Kook SH. The Long-lasting Radioprotective Effect of Caffeic Acid in Mice Exposed to Total Body Irradiation by Modulating Reactive Oxygen Species Generation and Hematopoietic Stem Cell Senescence-Accompanied Long-term Residual Bone Marrow Injury. Aging Dis. 2019 Dec 1;10(6):1320-1327. doi: 10.14336/AD.2019.0208.
3. Zhang H, Zhai Z, Wang Y, Zhang J, Wu H, Wang Y, Li C, Li D, Lu L, Wang X, Chang J, Hou Q, Ju Z, Zhou D, Meng A. Resveratrol ameliorates ionizing irradiation-induced long-term hematopoietic stem cell injury in mice. Free Radic Biol Med. 2013 Jan;54:40-50. doi: 10.1016/j.freeradbiomed.2012.10.530.
4. Xu G, Wu H, Zhang J, Li D, Wang Y, Wang Y, Zhang H, Lu L, Li C, Huang S, Xing Y, Zhou D, Meng A. Metformin ameliorates ionizing irradiation-induced long-term hematopoietic stem cell injury in mice. Free Radic Biol Med. 2015 Oct;87:15-25. doi: 10.1016/j.freeradbiomed.2015.05.045.
We acknowledge the reviewer’s comments and suggestions very much, which are valuable in improving the quality of our manuscript. Thank you and all the reviewers for the kind advice.
Sincerely yours.

Reviewer 2 Report
1. The manuscript is capacious. So much data, so many experiments, so many mice sacrificed.
2. The subject of the paper is on top in the field of medical answer for radiation event or to mitigate side effect of radiotherapy.
3. Everything looks nice: perfect English, concise description of Materials and Methods and Results. Only 2 sentence’ Conclusions. Even though the paper is impressive and to huge to present rough data. The figures are prepared for falcon sight people, e.g. Fig. 3 B or Fig. 5 A -there is no option to read the units of X axe. Generally figures are too tiny and their colours do not fit to black and white reprints.
4. There are dozen of acronyms in the text. Some of them not explained at all, or I couldn’t find explanation. The result is that the list of acronyms is necessary.
5. Under almost each of Fig. there are explanation of asterisks: * - P<0.05, ** - P<0.05 and *** - P<0.05. Why to have different designation for the same level of significances?
6. Line 102: the blood was taken several times during 2 months not 2 months after irradiation exactly.
7. Lines 263 – 264 mice show greater colony numbers? Have you investigated colonies of mice? I would change the expression somehow.
8. As You see my hints are in fact minor and do not change my impression that You did great job.
9. In Conclusions should be mentioned somehow that the experiment were conducted on mice not human.
Author Response
6th February, 2023
Professors
Section Managing Editor
Molecules
Manuscript ID: molecules-2174091
Manuscript Title: rhTPO ameliorates radiation-induced long-term hematopoietic stem cell injury in mice
Dear Editor,
Thank you for your letter in 21 January, 2023. On behalf of my co-authors, we thank you very much for giving us an opportunity to revise our manuscript, we appreciate editor and reviewers very much for their positive and constructive comments and suggestions on our manuscript.
Appended to this letter is our point-by-point response to the comments raised by the reviewer. The comments are reproduced and our responses are given directly afterward in a different color(highlight yellow).
We would like to express our great appreciation to you and reviewers for comments on our paper. We hope that we have responded satisfactorily to the constructive comments of the reviewers, and the paper will now be suitable for publication in the Molecules.
Best wishes,
The authors of paper molecules-2174091.
Response to Reviewer 2 comments
Thank you for your letter and for the reviewers’ comments concerning our manuscript entitled “rhTPO ameliorates radiation-induced long-term hematopoietic stem cell injury in mice” (Manuscript ID: molecules-2174091). Those comments are all valuable and very helpful for revising and improving our paper, as well as the important guiding significance to our researches. We have studied comments carefully and have made correction which we hope meet with approval. Revised portion are marked in highlight yellow(normal revision) in the paper. The main corrections in the paper and the responds to the reviewer’s comments are as flowing:
Point 1: The manuscript is capacious. So much data, so many experiments, so many mice sacrificed.
Response 1: Thanks for this comment. Moreover, we sincerely hope that our paper could meet the standards for publication here.
Point 2: The subject of the paper is on top in the field of medical answer for radiation event or to mitigate side effect of radiotherapy.
Response 2: Thanks for this positive comment.
Point 3: Everything looks nice: perfect English, concise description of Materials and Methods and Results. Only 2 sentence’ Conclusions. Even though the paper is impressive and to huge to present rough data. The figures are prepared for falcon sight people, e.g. Fig. 3 B or Fig. 5 A -there is no option to read the units of X axe. Generally figures are too tiny and their colours do not fit to black and white reprints.
Response 3: Thank you for this vital comment. We have revised the figure size and colors, and supplemented all units of figures.
Point 4: There are dozen of acronyms in the text. Some of them not explained at all, or I couldn’t find explanation. The result is that the list of acronyms is necessary.
Response 4: Thank you for your valuable advice. We have now added a list of acronyms to address more clearly (please see lines 447-453, page 14).
Point 5: Under almost each of Fig. there are explanation of asterisks: * - P<0.05, ** - P<0.05 and *** - P<0.05. Why to have different designation for the same level of significances?
Response 5: Thank you for pointing this important suggestion. We are very sorry for our sloppy work, and we have changed the explanation of asterisks.
Point 6: Line 102: the blood was taken several times during 2 months not 2 months after irradiation exactly.
Response 6: We are grateful for this important comment, and we have now modified the expression to address more clearly (please see lines 106-108, page 3).
Point 7: Lines 263 – 264 mice show greater colony numbers? Have you investigated colonies of mice? I would change the expression somehow.
Response 7: Thank you for your insightful comments. We have changed the expression to ”mice group showed greater BM-derived clonogenic activity” in lines 264-268, page 10.
Point 8: As You see my hints are in fact minor and do not change my impression that You did great job.
Response 8: Thank you for your comments.
Point 9: In Conclusions should be mentioned somehow that the experiment were conducted on mice not human.
Response 9: Thank you for underlining this deficiency. We are sorry for the confusion and now corrected (please see lines 426-429, page 13-14).
We acknowledge the reviewer’s comments and suggestions very much, which are valuable in improving the quality of our manuscript. Thank you and all the reviewers for the kind advice.
Sincerely yours.

Round 2
Reviewer 1 Report
All the comments by reviewer were addressed by the authors and improved the overall quality of the manuscript.